# Exploring concepts of compassion in care home settings: a scoping review protocol

Jay Evans ,[1,2] Elizabeth Grant ,[1] Anne Birgitta Pessi ,[3] Laura Evans ,[1] Silja Voolma [4]

¹Usher Institute of Population Health Sciences and Informatics, The University of Edinburgh, Edinburgh, UK
²University of Copenhagen, Kobenhavn, Denmark
³Faculty of Theology, University of Helsinki, Helsinki, Finland
⁴University of Tartu, Tartu, Estonia

**Correspondence to**
Jay Evans; Jay.Evans@ed.ac.uk

## ABSTRACT

**Introduction** There is widespread agreement that medical care without compassion cannot be patient-centred, but patients still routinely cite that they feel a lack of compassion in the care environment. There is a dearth of research on how compassion is experienced in a non-hospital setting such as a care home, not just by residents but by staff and other key stakeholders. This scoping review aims to determine the body of existing, published research that explicitly refers to compassion or empathy in the context of care homes.

**Methods and analysis** This scoping review will follow the methodology described by Arksey and O'Malley and the PRISMAextension for scoping reviews guideline to adhere to an established methodological framework. Relevant publications will be searched on the EMBASE, MEDLINE, PubMed, CINAHL, EBM Reviews and PsycInfo databases. Peer-reviewed literature focusing on experiences of compassion or empathy in care home settings from the perspective of either staff, residents (or clients), family members or their combined perspectives will be included. We will focus on literature published from 2000 up to 1 November 2021, in English, Spanish, Portuguese Finnish and Estonian. The review process will consist of three stages: a title review to identify articles of interest, this will be followed by an abstract review and finally, a full-text review. These three stages will be conducted by two reviewers. Data will be extracted, collated and charted and a narrative synthesis of the results will be presented.

**Ethics and dissemination** Ethical approval is not required for this scoping review. This study supports the first part of a larger programme to understand the importance of technologies in care homes. The scoping review will examine data from publicly available documentation, reports and published papers. Dissemination will be achieved through engagement with stakeholder communities, and publishing results. Our team will include representatives from the different communities involved.

## Strengths and limitations of this study

► This will be one of the first scoping reviews that seek to explore the evidence base available around experiences of compassion in care home settings.

► This review will explore compassion from the perspective of patients, families and caregivers in non-hospital settings (care homes primarily) filling a void in the literature identified by previous studies.

► The search strategy includes six electronic databases, broad search terms, and the use of forward citation tracking to ensure wide coverage of the subject area.

► This scoping review is the first step in a series of studies seeking to further explore and understand compassion in the context of care homes, thus further widening the evidence base available.

► This scoping review will only include peer-reviewed literature to have a manageable number of studies to screen and analyse, potentially omitting some relevant evidence in the grey literature.

outcomes when a good connection between care provider and patient is present and improved job satisfaction for care providers.[2 3]

Although experiences of compassion may vary, some researchers have proposed a working definition of compassion that includes recognising suffering in others; understanding the common humanity of this suffering; feeling emotionally connected with the person who is suffering; tolerating difficult feelings that may arise; and acting or being motivated to act to help the person.[4] There are slight variations to this definition depending on the cultural environment of the care provider.[5 6] However, only recently a consensus has developed around a general definition of compassion within healthcare settings that is backed by empirical evidence.[7]

The definition offered by Sinclair *et al*[8] focuses on an evidence-based definition of compassion following multiple literature reviews and an in-depth study among palliative care patients. The paper concludes that

## INTRODUCTION

Among patients and care providers globally, there is agreement that medical care without compassion cannot be patient-centred. Compassionate care addresses patients' needs to connect and develop relationships.[1] Evidence also suggests better clinical

compassion, 'is a virtuous response that seeks to address the suffering and needs of a person through relational understanding and action'. The elements of this definition are supported by earlier studies involving older patients in acute care settings where researchers found that patients identified a compassionate response with those aspects of empathy and action as detailed in the definition offered by Sinclair *et al*.[9] For the purposes of this literature review, we will use the definition of compassion offered by Sinclair as our working definition. However, within the review, we will continue to examine the validity of the definition as well as discuss other concepts of compassion as they may arise during the review.

The consensus that has been built around a definition of compassion in healthcare settings has been complicated by a growing need to begin to measure compassion as an outcome of care.[10] This importance is further illustrated by examples of negligent care where compassion was notably absent.[11] However, to begin to measure a construct such as compassion we need to be able to examine how it is experienced across a variety of care settings as well as stakeholders.

There are still significant gaps in our understanding of how patients and families of patients perceive compassionate care within the care environment. Studies have shown that perceptions of compassion can vary widely between patient and physician.[12] Despite increased involvement of patients in studies examining compassionate care, there is still a shortage of empirical studies examining their perspectives.[13]

There remains a paucity of research on the needs of residents and staff in a care home setting. Patients routinely cite that they feel a lack of compassion in the care environment.[1 14 15] This could be the result of several factors including no fully evaluated interventions to deliver compassionate care, absence of fully validated tools to determine the level of compassion in the care environment, and no significant evaluation of how compassion is taught to key care providers such as nurses.[16]

Numerous articles have identified the need to further explore compassion across the continuum of care and from the perspective of a diverse group of stakeholders in the care pathway including patients and families.[17 18] Non-hospital settings also remain underrepresented in studies on compassion.[9] This includes care homes as well as care workers. Tackling the hurdles of effectively describing factors that impact compassionate care within the care home environment would also create an enabling environment for the development of a curriculum for health workers/care workers that is focused on teaching strategies to address the challenges of delivery of compassionate care in the face of a dramatic increase in demand for care home space and care workers across the whole of the UK and globally.[19]

Staff experience stress, including compassion fatigue, at the same levels as full-time nursing staff in a hospital setting.[20] These combined factors create an acute need to examine the extant literature around how compassion is experienced in a non-hospital setting such as a care home, not by just the residents of the home but by staff and other key stakeholders.

A scoping review was identified as an effective way to examine the existing body of literature as it provided the needed flexibility to examine a wide range of sources as well as what will undoubtedly be significant heterogeneity concerning the methods used in the articles selected. This scoping review will follow the methodology and framework described by Arksey and O'Malley.[21] The team has also determined to use the Preferred Reporting Items for Systematic Reviews and Meta-Analyses (PRISMA) extension for scoping reviews as the governing structure for the review. It was agreed by the review team that the PRISMA extension would allow for reproduction of the results by other teams of reviewers as well as provide transparency to the review process.[22]

This scoping review aims to determine the body of existing, published research that explicitly refers to compassion or empathy in the context of care homes. The specific objectives are:

► To clarify the evidence base available around experiences of compassion in care home settings. Clarification will be made by a review of the evidence base of journals and abstracts in this topic area, looking at all designs of study.
► To examine how the concept of compassion is defined within the extant evidence base and examine whether there is consensus around the definition.

## METHODS AND ANALYSIS

To best achieve the objectives of the review the team have carefully considered the review methods and strategy for analysis. The review process will consist of three stages: first, a title review to identify articles of interest, this will be followed by an abstract review and finally, a full-text review. For the first two stages of screening, the titles and abstracts of articles retrieved in the search will be read and analysed by two members of the review team to identify potentially eligible articles. Emphasis will be placed on excluding studies that clearly do not meet the inclusion criteria cited above based on their title. Subsequently, study abstracts will be examined to determine if they meet the SPICE criteria defined below. Studies clearly not adhering to the inclusion criteria will be removed. Any disagreements on inclusion/exclusion of articles reviewed for title and abstract will be brought to a third reviewer for final determination for inclusion/exclusion.

Afterwards, two team members (Jay Evans and Anna Birgitta Pessi) will then each assess the full-text articles to determine whether they meet the inclusion/exclusion criteria. Any disagreement regarding the full-text articles to be included will be reviewed a second time, and further disagreements about study eligibility at the full-text review stage will be resolved through discussion with a third investigator (Elizabeth Grant) until the team can reach an agreement regarding the article in question. Any articles identified by 'snowballing' and forward citation

tracking, as mentioned earlier, will be subject to the same set of review criteria.

The review will not establish restrictions regarding geographic locations, although distinctions between studies conducted in higher-middle income countries and lower-middle income countries will be made when applicable. Only studies written in the five languages specified above will be included. The review will not involve the public or patients in its design, conduct, reporting or dissemination as it is not common practice when carrying out scoping reviews.

We will describe key categories, such as the target populations, intervention characteristics and types of questions posed. This review of the literature will also provide suggestions for future research. Potential gaps will also be identified. The data collected will be stored in an electronic database in MS Excel and the results of the rapid review will be presented descriptively in tables and graphs.

Data extracted for analysis within the abstract and articles will be managed via Covidence software (web).[23] The references for the review will be maintained using an electronic reference management software.

Narrative synthesis of the included articles will be carried out using a framework that consists of three elements: (1) developing a preliminary synthesis of findings of included studies; (2) exploring relationships within and between studies; (3) assessing the robustness of the synthesis.

To maintain the integrity of the review we will use the PRISMA extension for scoping reviews checklist to double-check the team's adherence to proper review strategy and protocol.

### Setting, Perspective, Intervention, Comparisons, Evaluation

We decided on the use of the SPICE (Setting, Perspective, Intervention, Comparisons, Evaluation) framework in order to structure the approach to our search.[24] We felt the SPICE structure allowed for examination of our population of interest from two important approaches: setting and perspective. The review team believes that this will allow for a more structured search given the complexity of examining concepts such as compassion within care home settings. We also felt that Evaluation was a more appropriate frame than Outcome that is used in the PICO (Patient Problem, Intervention, Comparison, Outcome) framework.[25]

### Setting

Care homes. In particular care homes providing services for older persons (persons over 65 years of age). We are interested in looking at care homes not just in the UK but around the world.

### Perspective

We are interested in the experiences of compassion from the perspective of residents (or clients), staff and family members of residents who are in a care home.

### Intervention

People who have been admitted to a care home or working in a care home on a long-term basis whether in a full-time capacity or part-time.

### Comparisons

Residents/staff in other similar care facilities such as daycare.

### Evaluation

We wish to examine experiences of compassion of people (residents, staff, family members of residents) in a care home setting.

### Inclusion criteria

► Peer-reviewed literature focusing on experiences of compassion or empathy in care home settings.
► Studies that focus on the experience of compassion or empathy in care homes from the perspective of either staff, residents (or clients), or family members, or from combined perspectives.
► Studies published in English, Spanish, Portuguese, Finnish and Estonian.
► Literature published from 2000 onwards. (The review team decided that due to significant changes within the care sector over the last two decades, literature published over the last 20 years would likely yield the most appropriate results in order to achieve the objectives of the review.)

### Exclusion criteria

► Literature not published in either English, Spanish, Portuguese, Finnish and Estonian as the review team do not have the ability to properly evaluate other languages outside of the five listed.
► Literature that is not peer-reviewed.
► Studies that have not included participants in a care home setting.
► Literature published before 2000.
► Studies that focus on other related concepts (eg, compassion fatigue, self-compassion, caring, ethics, communication) or use interventions that aim to foster self-compassion (eg, mindfulness-based stress reduction, compassion-focused psychotherapy).
► Studies focusing on acute care settings such as a hospital.
► Articles without the full text available for review.

### Search strategy

The review team will comprise both content and methodological experts. Searches of electronic databases between 15 November 2021 and 31 December 2021, will be conducted, including EMBASE, MEDLINE, PubMed, CINAHL, EBM Reviews and PsycInfo. We will focus on literature published from 2000 up to 1 November 2021. The term compassion is employed in the healthcare literature in a variety of ways and its relationship to similar concepts such as empathy is sometimes not well defined.[7] Therefore, we will keep the search terms broad to ensure wide coverage of the subject area. We

will also screen all reference lists of the included studies to identify additional studies of relevance to be screened. Via the use of 'snowballing' as mentioned, we hope to capture additional studies that were not found in the initial searches to further strengthen the rigour of the scoping review.[17] We will also use forward citation tracking as well.

The search terms of compassion, empathy, sympathy and caring will be combined with medical subject headings (headings, subheadings, publication types) along with appropriate wildcard terms that may include: care home, residential care, elder care, elder care home, elderly care and so on. Search terms will be combined with the appropriate Boolean operator, such as 'AND', 'OR' to achieve the refinement needed in the search results. The team will also conduct a search of other relevant sites such as the WHO and the National Health Service. The search strategy will be reviewed and validated by the team members in consultation with the University of Edinburgh library staff. Any changes to the search strategy based on recommendations will be noted and terms/strategy will be altered accordingly.

### Patient and public involvement
No patient involved.

## ETHICS AND DISSEMINATION
Ethical approval is not required for this scoping review. This study supports the first part of a larger programme titled, 'Technology's impact on compassion within care homes: A study of resident, family and staff perspectives of the impact of technology in care home settings' supported by the Economic and Social Research Council to understand the importance of technologies in care homes, particularly in relation to compassion. The scoping review will review data from publicly available documentation, reports and published papers. Dissemination will be achieved through engagement with stakeholder communities, and publishing results. Our team will include representatives from the different communities involved.

**Contributors** JE (corresponding author) authored the framework and lead the development of the protocol. EG contributed to the section on methods and review of the overall approach to the search strategy. ABP contributed significant narrative around how compassion is defined and how to best approach the subject within the context of the protocol. LE provided support for proofreading, formatting and referencing for the protocol as well as contributing to the overall writing. SV provided additional insights and content in the search strategy and protocol development.

**Funding** This work was supported by the University of Edinburgh's ESRC Impact Accelerator Grant scheme grant number ES/T50189X/1.

**Competing interests** None declared.

**Patient and public involvement** Patients and/or the public were not involved in the design, or conduct, or reporting, or dissemination plans of this research.

**Patient consent for publication** Not applicable.

**Provenance and peer review** Not commissioned; externally peer reviewed.

**ORCID iDs**
Jay Evans https://orcid.org/0000-0002-4948-6027
Elizabeth Grant https://orcid.org/0000-0001-7248-7792
Anne Birgitta Pessi https://orcid.org/0000-0002-1312-9538
Laura Evans http://orcid.org/0000-0002-6187-6298
Silja Voolma https://orcid.org/0000-0002-5496-2614

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
