## [Reviewer comments · BMJ Open]

ARTICLE DETAILS

TITLE (PROVISIONAL)	Exploring concepts of compassion in care home settings: A scoping review protocol
AUTHORS	Evans, Jay; Grant, Elizabeth; Pessi, Anne; Evans, Laura; Voolma, Silja

VERSION 1 – REVIEW

REVIEWER	Heward, Michelle Bournemouth University
REVIEW RETURNED	05-Aug-2021

GENERAL COMMENTS	Thank you for undertaking this worthwhile scoping review. I think the protocol is clear and well written, but that you could clarify the following to improve the paper: 1. Please change all references to 'elderly people' to 'older people' as this terms is more widely accepted now.2. Will the two team members undertaking the review be the same two throughout the process or not - perhaps you could add the initials in brackets to clarify who will be undertaking this review and who will be acting as the third reviewer when needed?3. You mention in the ethics and dissemination section that this review is 'the first part of a larger programmer to understand the importance of technologies in care homes'. As a reader I am not clear what this wider piece of work is and how this review links to it, please can you provide more information? Good luck with your review.
---

REVIEWER	Webster, Lucy UCL, Division of Psychiatry
REVIEW RETURNED	06-Sep-2021

GENERAL COMMENTS	Overall, this is a clearly written protocol for this scoping review. Minor point but it would be helpful if the Comparisons element in the SPICE criteria is more clearly defined.
--

VERSION 1 – AUTHOR RESPONSE

Reviewer: 1

Dr. Michelle Heward, Bournemouth University

1. Please change all references to 'elderly people' to 'older people' as this term is more widely accepted now.

Response: Thank you for this valuable recommendation. The terms within the text have been changed accordingly.

2. Will the two team members undertaking the review be the same two throughout the process or not - perhaps you could add the initials in brackets to clarify who will be undertaking this review and who will be acting as the third reviewer when needed?

Response: Again, thank you for this helpful note. The responsibilities for each team member have been included within the footnotes to indicate their respective roles in the review process.

3. You mention in the ethics and dissemination section that this review is 'the first part of a larger programme to understand the importance of technologies in care homes'. As a reader I am not clear what this wider piece of work is and how this review links to it, please can you provide more information?

Response: Thank you for this note. We have added text to explain the link to a larger study that this review complements.